# Study on Imagery Modeling of Solid Wood Chairs in Big Data

Le Xu [1,2] and Younghwan Pan [2,*]

1 College of Design, Zhijiang College of Zhejiang University of Technology, Shaoxing 312030, China; xule0297@kookmin.ac.kr

2 Department of Smart Experience Design, Kookmin University, Seoul 02707, Republic of Korea

\* Correspondence: peterpan@kookmin.ac.kr

**Abstract:** With the continuous improvement of living quality and aesthetics, people have increasingly higher requirements for furniture products. Excellent solid wood chairs are one of the most representative products in the furniture industry. To enhance space and taste, the design of chairs may significantly impact consumers' emotional experiences and purchase decisions. This study aims to evaluate how the modeling imagery of solid wood chairs affects consumers' preferences and emotional experiences. The development of the current era is inseparable from the analysis of big data. Firstly, a representative sample is obtained by multidimensional scaling (MDS), analyzed, and evaluated by factor analysis. Moreover, five groups of adjective vocabulary are selected to describe the modeling imagery of solid wood chairs, such as "balanced and coordinated", "unique and novel", "practical and simple", "quality and detailed", and "traditional and plain". Further, the triangular fuzzy theory is applied to analyze and discuss the twelve types of solid wood chairs in the five groups of adjective vocabulary. Then, the study verifies that the differences are significant in the evaluations of the 12 samples in the "unique and novel" and "quality and detailed" groups, and small in the groups of "traditional and plain", "balanced and coordinated", and "practical and simple". Through comprehensive comparisons, five groups with similar modeling imagery are created, and solid wood chairs with different modeling imagery should be placed in suitable spaces. According to the results of this study, the evaluation of the modeling imagery of solid wood chairs cannot solely rely on subjective judgments. However, it can be reasonably refined through data analysis and mathematical algorithms. It can also scientifically and effectively reflect the potential perception needs of consumers on the modeling imagery of solid wood chairs, as well as help to improve the design efficiency of the furniture product development stage.

**Keywords:** furniture design; solid wood chairs; multidimensional scaling; Kansei engineering; modeling imagery; factor analysis; triangular fuzzy theory





## 1. Introduction

With the rapid development of education and the economy, people's living standards and the public's aesthetic have dramatically improved. Under these circumstances, people have begun to pay attention to the quality of life. When purchasing products, they pay more attention to the aesthetics of the products' shape [1]. Furniture is an essential living appliance for human life. Furniture is inseparable from people's clothes, food, housing, and transportation; it contains positive aesthetic meaning and rich cultural connotations. Improving the human living environment and quality of life is also meaningful.

With the relaxation of COVID-19 restrictions worldwide, the furniture market will gradually grow in operating revenue. According to Statista data, the operating revenue of the furniture market is expected to reach $809 billion by 2024. The market is expected to annually grow at 5.31% (CAGR from 2023 to 2027) [2]. Due to the continuous advancement in processing technology, the materials used in furniture are becoming increasingly more abundant, which can be roughly classified into wood, metal, plastic, etc. [3]. In a survey of the global furniture market in 2021, wood furniture accounts for about 40%,

occupying the critical market share [4]. In addition, wood furniture is green and beneficial to people's health compared to materials that require the addition of various chemicals and processing [5], so consumers have a particular preference for wood furniture. With the rapid economic development of the past two decades, China has become one of the world's largest furniture producers. The output of China's furniture industry in 2020 was 912.21 million pieces, an increase of 1.7% year-on-year [6]. The market scale of wooden furniture has been expanding, and the furniture industry is also on the rise [7], which strongly represents China's furniture industry.

According to sociological analysis, most people spend more than two-thirds of their lives in contact with furniture. In addition to the bed, the chair has the longest contact time and the most closely related items with people. Hence, the solid wood chair is the most competitive in the market and the favorite furniture among consumers. There are various types of chairs, which can be divided into office, dining, bar, leisure, reclining, children's, etc., according to different usage. Solid wood chairs are generally used as dining chairs, lounge chairs, and recliners due to the characteristics of the material, structure, and craftsmanship.

As a source of innovation, design has recently received widespread attention from researchers and practitioners. Increasingly more furniture manufacturers regard furniture design as an essential strategic asset and core competitiveness [8]. Whether from ancient to modern times, or from the East to the West, the chair has been an eternal theme in furniture design [9]. Chair design involves various factors such as shaping, function, material, structure, technology, aesthetics, and ergonomics, with a systematic theoretical basis and rich practical experience. However, it is also the best embodiment of the designer's concept and design level. The chair is a product that highly integrates practicality and art. In addition to meeting the consumers' functional needs of sitting and resting, designers must consider consumers' unique needs in the intellectual, emotional, aesthetic, and cultural aspects of spirituality [10]. The names of the various chair parts, such as the legs, back, surface, and armrests, are similar to human body parts and somehow like humanized furniture. A good chair design even has its own personality. The chair has a self-contained and complete sense of beauty, the beauty of its shape is no less than sculpture, and it also conveys the spirit of culture with the power to dominate the space. Hans J. Wegner, a Danish furniture designer, once said: "Imagine if you could design just one good chair in your lifetime—but that simply cannot be done". It means that chair design is a very challenging task for furniture designers.

Looking back at history, countless classic chair designs emerged. Finnish designer IImari Tapiovaara believes that the chair's design is the beginning of any interior design. For example, architects should design chairs for their buildings; interior designers should consider chairs in combination with space, and product designers also explore new materials to design chairs. Today, there are increasingly more furniture brands on the market, and the styles of solid wood chairs are also diverse. The styling characteristics of chairs are more easily perceived and distinguished by users than ergonomic features [11]. In addition, different styling attributes can cause different emotional experiences, influencing the consumers' preference for styling [12]. Designers usually work based on their experience and aesthetics in developing and designing solid wood chairs. Because consumers and designers live in different environments and have different needs, then their perception of the product shape varies, which directly affects the consumers' preference for solid wood chairs.

In summary, it is meaningful to objectively analyze and study the relationship between different shapes of solid wood chairs and people's visual and psychological factors, and to explore consumers' evaluation of various solid wood chairs. On the one hand, it is beneficial to narrow the gap between designers and consumers regarding the perception of solid wood chair modeling; on the other hand, it is also conducive to improving the design value of solid wood chairs for furniture manufacturers, designers, and consumers. Existing solid wood chair size proportion, color, and material commonly affect consumers'

preference for solid wood chair modeling. Thus, solid wood chair modeling design in the whole design process is essential. Designers should focus on meeting the potential emotional needs of consumers and carrying out design development work to present a better user experience of solid wood chairs.

People's dining, office, reading, resting, and waiting lifestyles are inseparable from the chair, especially the solid wood chair. It is an integral part of our homes. Mckellar's five-sensory survey of images found that visual imagery has the highest proportion [13]. However, modeling is the most intuitive presentation of visual imagery. In addition, product "modeling" characteristics are the most effective in consumer perception, which shows that the aesthetics of product modeling is one of the most critical factors influencing consumers' purchasing and use, and the aesthetics of product modeling is one of the most critical factors influencing this [14,15]. A study exploring the aesthetic characteristics of products with the most significant impact on consumers' emotional pleasure showed that styling factors could directly and effectively influence consumers' perceptions of emotions. However, emotions are also an essential factor influencing customers' purchasing behavior [16]. First, many researchers have considered solving the problem of the best sitting posture. They have studied ergonomics in-depth and use relevant information about the human body when making chairs to make them more comfortable [17]. Then, some researchers have tried to design chairs by opening a pneumatic seating system (PASS), a new human-computer interaction tool [18]. Furthermore, some researchers have used computer-aided and finite element multifunctional analysis to design chairs and analyze their structures to effectively assess the furniture's strength [19,20]. Other researchers have applied shape syntax and parametric methods for chair design to enhance the efficiency of the mass customization of chairs [21]. With the rapid development of artificial intelligence, researchers have also learned to use convolutional neural networks and generative adversarial networks to assist chair design in improving design efficiency [22,23]. However, most of the existing research on chair design has focused on the functional and technical aspects, and they need to place more emphasis onto the connection between solid wood chair modeling and consumers' emotion. In summary, furniture is a symbol of the level of social productivity development of a country or region in a particular historical period; sitting furniture is the most significant sales volume of a category, of which the solid wood chair is essential to single product furniture. In terms of modeling design, visual presentation, and modeling imagery perception, relatively little research on solid wood chairs has been conducted. The use of perceptual engineering and fuzzy theory to study the modeling imagery of solid wood chairs is an effective method, which is conducive to enterprises and designers quickly and accurately grasping the consumer and market demands. This research result is more scientific and practical, because a representative sample was chosen using a scientific sample selection technique called multidimensional scaling (MDS). Because a two-dimensional image of the sample was redrawn to eliminate the interference of other factors like angles, colors, and materials, the results of this study can be considered to be scientific and practical.

This study used objective methods, such as perceptual engineering and fuzzy theory, to evaluate and study consumers' visual and psychological impressions of solid wood chairs with different shapes. The goal was to learn about customers' buying intentions and preferences. First, this study attempted to transform consumers' feelings and intentions into design elements and provide a reference for modeling imagery and styles when designing solid wooden chairs. The designers could then design products most suitable for consumers and market needs. Secondly, it could enable furniture manufacturers to position brands and furniture styles more accurately according to the characteristics of target users. Thirdly, the modeling imagery research helps analyze consumers' consumption intention regarding the brand and the appearance of the solid wood chairs. Consumers can choose the right solid wood chair according to their home decoration style and personal preferences, which can provide practical style references for consumers and improve their quality of living. It

could further promote the consumer to solid wood chair modeling imagery potential and perceived demand research.

## 2. Literature Review

Since 1980, many studies have focused on the computational and experimental testing of chair structures, and researchers have mainly investigated chair-related calculations. Some researchers have performed numerical calculations using a "linear elastic model" of orthotropic materials to analyze the stiffness of a statically indeterminate wooden chair side frame, where the stiffness of the joints in the frame significantly affects the structural strength of the chair [24]. Some researchers have tested the mechanical properties of chairs by finite element analysis (FEM) to understand their load limit capacity (e.g., at high user weight) [25,26], and also by investigating the effect of the user's weight on the load capacity and size of the structural elements of the chair to design an optimally sized chair structure [27]. There have also been studies by numerical and experimental methods to compare the mechanical properties of two cantilevered leg connection structures of solid wood chairs to optimize the furniture structure [28]. In addition, some researchers have used the finite element method (FEM) and MATLAB nonlinear programming method for optimizing the frame structure of chairs [29], and some researchers have measured the human dimensions of tabloid chairs through experimental methods to make the chairs more ergonomic and comfortable to sit on [30]. Finally, five principles of the Alexander Technique have been applied to solve chair design problems [31]. All these study results influence the chair design's modeling properties. At the same time, although the current research on chairs covers all aspects, there needs to be more research based on objective research methods to explore the influence of solid wood chair modeling imagery on consumers' emotional experience and favorite preferences. Evaluating the modeling imagery of solid wood chairs based on perceptual engineering and fuzzy theory through questionnaires can help close the gap between designers' and consumers' perceptions of solid wood chair modeling and improve product development efficiency for furniture companies and designers.

### 2.1. Kansei Engineering

Kansei Engineering was founded by M. Nagamachi at Hiroshima University around 30 years ago and has spread worldwide. Perceptual engineering aims to develop new products by translating customers' psychological needs and feelings (sensibility) into design specifications [32]. The following four steps of perceptual engineering research [33] are used: (1) Product sample collection—a specific product type is selected, and sample images are collected. (2) Perceptual adjective collection and screening—collect the adjective vocabulary suitable for related design imagery. (3) Screening representative samples— invite experts in the field to conduct experiments and use MDS and cluster analysis to screen out representative samples. (4) Screening representative perceptual adjectives—combine the perceptual vocabulary from the second step with the representative sample from the third step and create an evaluation questionnaire. The data obtained from the experiment are analyzed by factor analysis and cluster analysis through SPSS software to extract several pairs of words regarding the perceptual imagery of the product. (5) Establishment of imagery space—a questionnaire comprised of the representative samples and the representative perceptual words obtained from the previous experiments. The respondents are invited to rate each sample for the representative adjectives. (6) Analysis of experimental data—the data obtained from the experiment are subjected to multiple regression analysis and triangular fuzzy theory analysis.

### 2.2. Multidimensional Scaling

Multidimensional scaling (MDS) is a technique for multivariate data analysis with reduced dimensionality. MDS transforms a set of points in a high-dimensional space into a low-dimensional space while maintaining the relative distance between pairs of points [34].

In other words, MDS aims to create a map of objects only described by a proximity matrix (similarity or dissimilarity). It is the representation of objects in a low-dimensional space (usually Euclidean) so that the distances between points reflect dissimilarity in some sense: similar objects are mapped as close to each other. In contrast, different objects are represented as points far from each other [35,36]. When choosing the optimal number of dimensions, reference may be made to the minimum stress index in each dimensional number scheme. The stress index takes values from 0 to 1. The lower the stress index, the more appropriate the dimensionality [37]. Regarding Kruskal's interpretation, the degree of matching at different stress indices is shown in Table 1.

**Table 1.** Kruskal stress index interpretation.

| Stress | Matching Degree |
|---|---|
| 0.200 | Poor |
| 0.100 | Fair |
| 0.050 | Good |
| 0.250 | Excellent |
| 0.000 | Perfect |

### 2.3. Factor Analysis

A statistical technique frequently used to streamline complex data is factor analysis (FA) [38]. Its main goal is to replace the original data structure with fewer dimensions while still holding most data [39]. The features of each dimension are identified by assessing the commonality and content of the data in each dimension. These factors can significantly reduce the overall complexity of a vast amount of data by replacing the original variables [40,41]. Kaiser's study found that factor analysis is more useful when the KMO value is closer to 1, which indicates a higher degree of variance correlation [42].

### 2.4. Fuzzy Theory

Fuzzy theory is a scientific approach for analyzing and processing information and data. It was first presented by the University of California's L.A. Zadeh, a control theorist mainly focusing on fuzzy set theory, fuzzy logic, fuzzy reasoning, and fuzzy control. In order to cope with imprecise and fuzzy data and to resolve decision issues in fuzzy settings, this procedure is computed using rigorous mathematical approaches [43–45]. According to Zadeh, our thoughts, logic, and perception of the world are inherently hazy. As a result, it is vital to use logical and fuzzy notions to express the matrix's priority level. The subject's internal feelings may be quantified using membership functions by converting them into values between 0 and 1, where the values represent potential influences and their corresponding consequences.

One of the most effective research techniques is fuzzy theories regarding meaning measurement. The fuzzy theory is used in design research today [46,47]. Fuzzy numbers are commonly used in these investigations; the three standard varieties are triangular, trapezoidal, and ordinary, with triangular fuzzy numbers being the most prevalent. The membership function of the triangular fuzzy number is what makes it unique. A triangle-shaped probability distribution is depicted [48]. Suppose an int function $\mu_i(x)$ of a triangular fuzzy number, which is t = $(t_1, t_2, t_3)$. When $t_1, t_2, t_3$ are real numbers and $t_1 \leq t_2 \leq t_3$. This membership function is shown in Figure 1 [49].

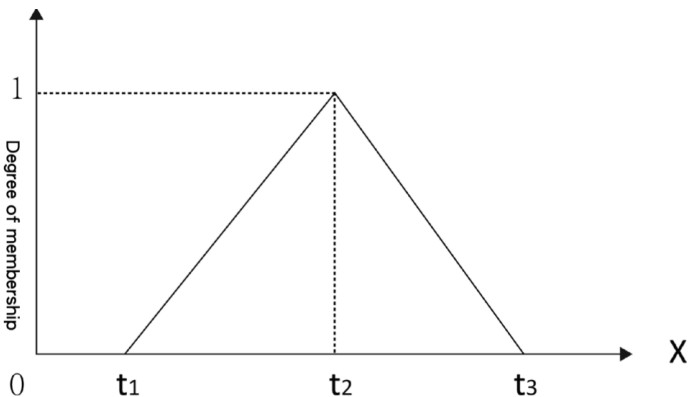

**Figure 1.** Membership function of a triangular fuzzy number.

The ranking approach in this study employed seven levels with a vague definition (see Table 2). Ranking a representative sample was requested from the public. A seven-level triangle membership function was used to obtain the fuzzy ranking results, which led to triangular fuzzy numbers following the quantification of the fuzzy meanings. Figure 2 depicts the triangular membership function.

**Table 2.** Linguistic variables for the importance and the ratings.

| Linguistic Variables | Triangular Fuzzy Numbers |
|---|---|
| Very Low (VL) | (0, 0, 1) |
| Low (L) | (0, 1, 3) |
| Medium Low (L) | (1, 3, 5) |
| Medium (M) | (3, 5, 7) |
| Medium high (MH) | (5, 7, 9) |
| High (H) | (7, 9, 10) |
| Very High (VH) | (9, 10, 10) |

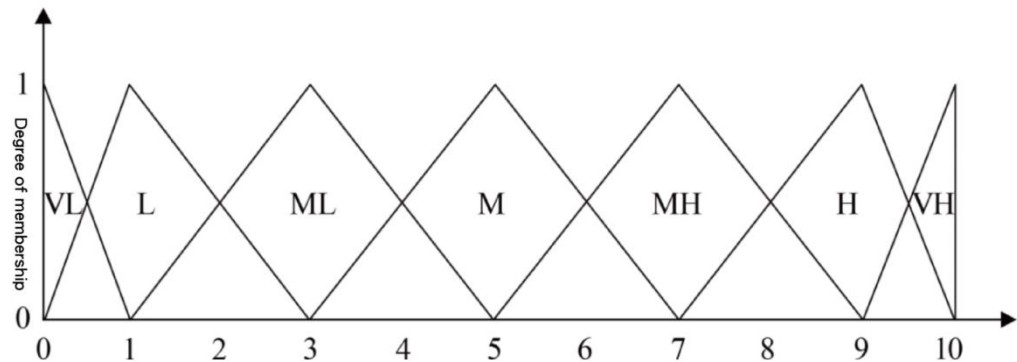

**Figure 2.** Membership function of a triangular fuzzy number.

The membership function demulsifies the triangle fuzzy numbers to obtain crisp values for further analysis and comparison. The most popular defuzzification technique employed in this work is the maximizing set and minimizing set method Chen introduced in 1985 [50]. The basic idea is to combine the weights of two triangular fuzzy integers to obtain the utility value overall. The specific calculation method is as follows:

Presume n numbers of the triangular fuzzy number in a membership function, which was defined as $\tilde{t}_i = (\tilde{t}_{i1}, \tilde{t}_{i2}, \tilde{t}_{i3})$, i = 1, 2,..., n; then, the minimum $\mu_G(x)$ and maximum $\mu_M(x)$ are G and M, respectively, as shown in Equation (1).

$$UT\left(\tilde{t}_i\right) = \frac{\left[\frac{\left(\tilde{t}_{i3} - X_{min}\right)}{\left((X_{max} - X_{min}) + \left(\tilde{t}_{i3} - \tilde{t}_{i2}\right)\right)} + 1 - \frac{\left(X_{max} - \tilde{t}_{i1}\right)}{\left((X_{max} - X_{min}) + \left(\tilde{t}_{i2} - \tilde{t}_{i1}\right)\right)}\right]}{2}, \; i = 1, 2, \ldots, n \quad (1)$$

### 2.5. Semi-Structured Interviews

As a data collection strategy, the semi-structured interview (SSI) has a unique structure that closely relates to qualitative, quantitative, and mixed-methods research. It can provide a comprehensive and accurate descriptive summary of participants' perspectives by converting textual information into data for analysis [51]. Furthermore, Michał Dolczewski (2022) proposed using semi-structured interviews as a data collection method to obtain more verbal and non-verbal material on self-esteem regulation [52]. However, fewer studies have used SSI methods to analyze product modeling imagery; therefore, this study used SSI to assess consumers' preferences and affective experiences of modeling imagery of solid wood chairs.

## 3. Research Methods

Based on the above theory, the framework of this study is shown in Figure 3. The practical steps are as follows: In the first step, the MDS method filtered a representative sample of solid wood chairs. An expert questionnaire was used to determine the range and number of adjectives describing the modeling of solid wood chairs. Factor analysis was used to determine the imagery adjectives that could represent the modeling of solid wood chairs. In the second step, the triangular fuzzy number calculation method was used to evaluate the adjectives describing the samples of solid wood chairs and to obtain the evaluation of imagery for the modeling characteristics of representative samples. The study's results were integrated and summarized in the third step to discuss the consumer's evaluation of representative solid wood chair modeling imagery. In the fourth step, the semi-structured interview data collection method provided consumers' favorite preferences and emotional experiences of the representative solid wood chair modeling imagery. This study investigated Chinese consumers' perceptions of the modeling imagery of solid wood chairs as the research object. It provides a reference for users to purchase furniture and designers in the furniture modeling design stage.

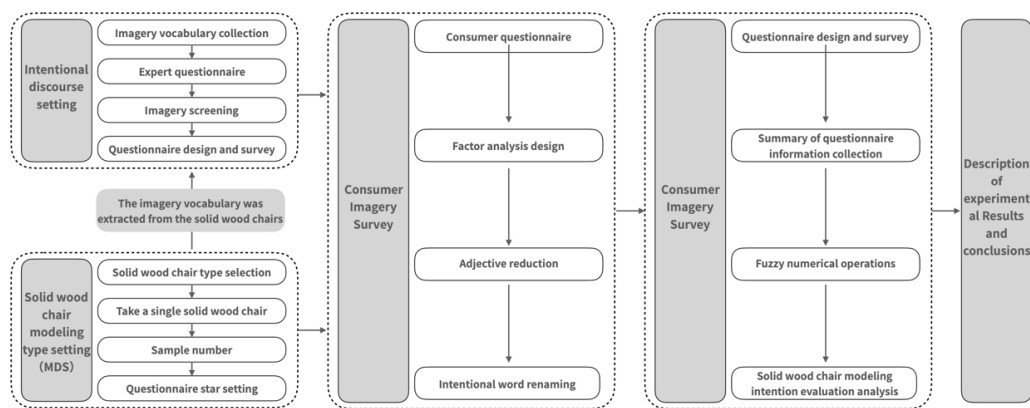

**Figure 3.** The research framework.

### 3.1. Step 1: Selecting Representative Samples

First, the samples of this study were taken from the official websites and furniture-sharing platforms of domestic and international furniture brands. Three hundred clear and

background interference-free images of three-quarters of solid wood chairs were collected as samples during this process. Due to the many samples, the solid wood chairs with poor quality and high similarity were removed through focal discussion and a pre-test by four design experts. Finally, 123 samples of solid wood chair shapes were selected. The remaining samples were coded using Adobe Photoshop; grayscale processing was performed, and the logo was removed to eliminate the impact caused by color and branding. Then, fifteen experts were invited to participate in the test (eight furniture design teachers working in a university, four furniture designers with over ten years of work experience, and three Ph.D. students majoring in design, shown in Figure 4). The physical features of the solid wood chairs in the sample were put into groups based on how similar they seemed to the participants (Figure 5 shows that the backrest, armrests, legs, cross braces 1 and 2, and chair surface were some of the morphological features). They created five to fifteen groups, allowing for the formation of a similarity matrix, which was later transformed into a similarity matrix for MDS analysis. The final, most significant samples were determined by clustering the samples using the results of the MDS as categorical variables.

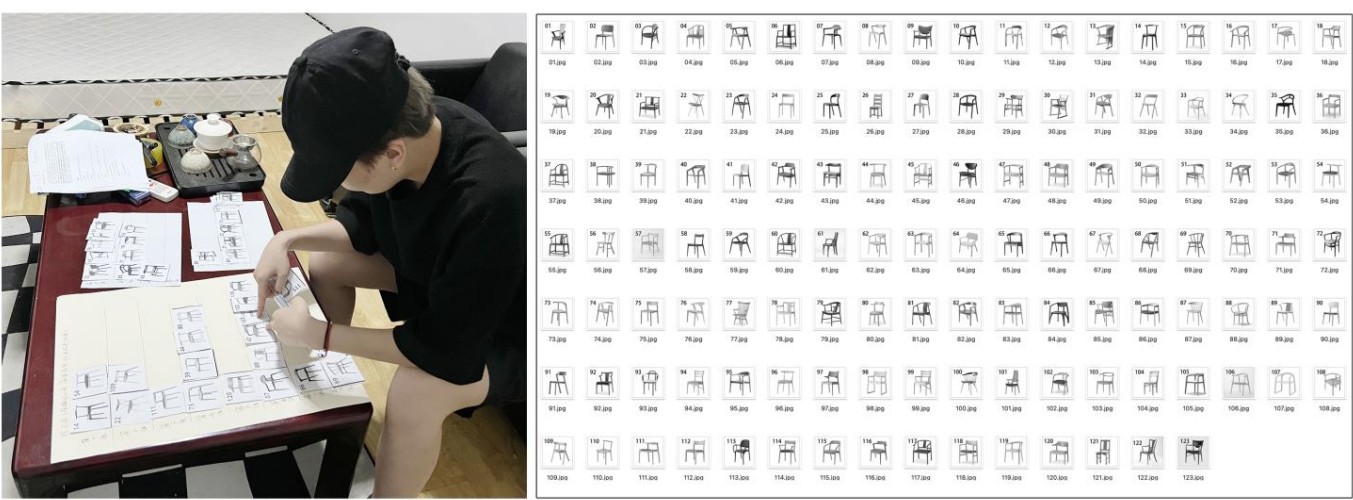

**Figure 4.** The participants in the test.

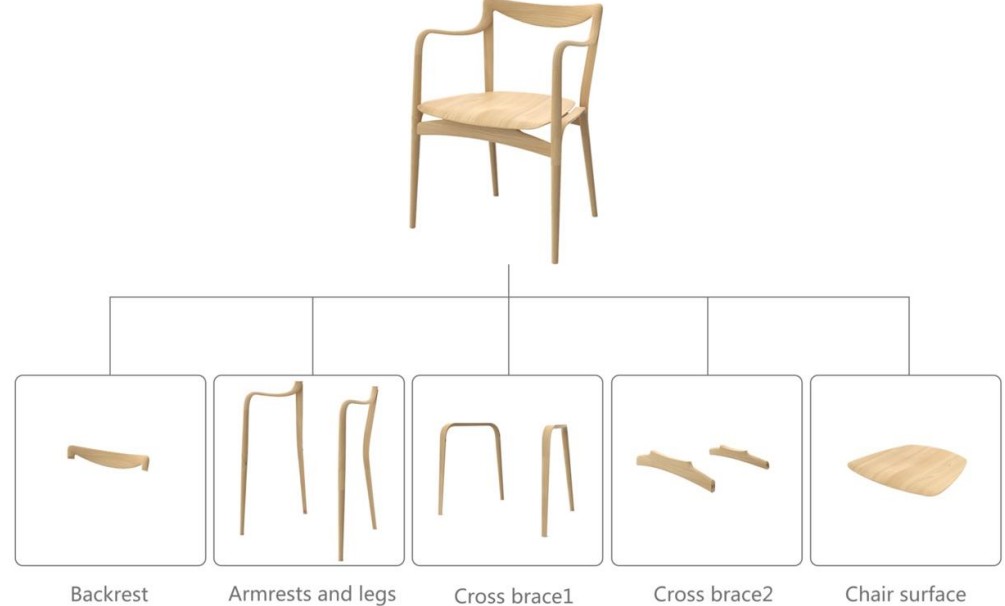

**Figure 5.** Composition of solid wood chair.

*3.2. Step 2: Extraction and Factor Analysis of Modeling Imagery Adjectives*

The words used to describe the modeling of solid wood chairs were taken from shared platform websites, company websites, and books on furniture. After subjective sorting, 140 adjectives were selected, and other questionnaires were administered to experts. The adjectives were categorized using the questionnaire method. Thirty-six people with experience designing furniture were asked to participate in the experiment. This included six company furniture designers, seven university design professors, fifteen postgraduate students, and eight Ph.D. candidates. Participants were asked to choose 40–50 words from a list of 140 words that best described how the chair was modeled. Then, based on how high or low each term was rated, the first 40 most recognizable adjectives were picked for more research. Briefly, factor analysis was employed to reduce the number of modeling imagery adjectives. At this time, 40 imagery adjectives had been collected, and the lexical variance had been determined by conducting a consumer survey using a five-point Likert scale. "Very inappropriate", "inappropriate", "moderate", "appropriate", and "very appropriate" were all included on the scale. One hundred twelve people filled out the surveys. We carried out factor analysis on the data, and the modeling imagery adjectives helped us find the right parts of the factors. The factors were finally given new names that reflected the traits of the modeling imagery adjectives found in the factor components.

*3.3. Step 3: Evaluation of Modeling Imagery in Fuzzy Theory*

The study's questionnaire was created utilizing the fuzzy-meaning approach. Twelve samples were paired with five adjective groups in the questionnaire, which used a seven-point Likert scale. For example, how do you rate the balance of coordination of sample S7 (S means sample)? How do you rate the uniqueness and novelty of sample S7? How do you rate the practical simplicity of sample S7? How do you rate the quality details of sample S7? How do you rate the traditional simplicity of sample S7? These questions helped determine consumers' opinions on the 12 solid wood chair samples. After fuzzing the results, the total utility values for each form of the modeling imagery were obtained, charted, and further analyzed for the 12 samples. These findings might serve as an inspiration for furniture companies, designers, and consumers.

*3.4. Step 4: Investigating Consumers' Preferences and Emotional Experiences of Modeling Imagery of Representative Samples*

This step used semi-structured interviews to investigate consumers' favorite preferences and emotional experiences of the modeling imagery of representative solid wood chairs. In this experiment, twenty participants (ten furniture designers and ten people with furniture buying experience) were invited to participate in the test through a convenience sample using semi-structured interviews, twelve of whom were male and eight female. The interview time lasted 20 min per person, and each respondent was given $7 as a reward, respectively. Finally, the data obtained from the experiment were further organized and analyzed.

## 4. Results and Discussion

*4.1. Results of Representative Sample Selection*

The study achieved the best results by MDS analysis with four dimensions (pressure = 0.08952), as shown in Table 3.

**Table 3.** The stress of different dimensions of MDS analysis.

| Dimensions | Stress |
|:---:|:---:|
| 2 | 0.15254 |
| 3 | 0.11235 |
| 4 | 0.08952 |

The samples were divided into 12 groups by cluster analysis using the MDS findings as a categorical variable. In comparison, the distance between each sample and the category's center was measured (see Table 4). The sample that is closest to the center is the one that best represents its category. Furthermore, we used Adobe Illustrator to turn these 12 exemplary examples into two-dimensional pictures of consistent style and angle to lessen the involvement of other elements and copyright concerns (see Figure 6). Figure 7 displays the final transformed 12 exemplary samples.

**Table 4.** The results of cluster analysis.

| Sample | Category | Distance | Sample | Category | Distance |
| --- | --- | --- | --- | --- | --- |
| S7 | 1 | 4.84651 | S26 | 7 | 2.90934 |
| S118 | 2 | 3.10012 | S36 | 8 | 3.51596 |
| S102 | 3 | 4.92846 | S43 | 9 | 2.47886 |
| S50 | 4 | 2.28655 | S12 | 10 | 2.76357 |
| S88 | 5 | 4.84236 | S76 | 11 | 4.80691 |
| S23 | 6 | 2.64236 | S63 | 12 | 4.18767 |

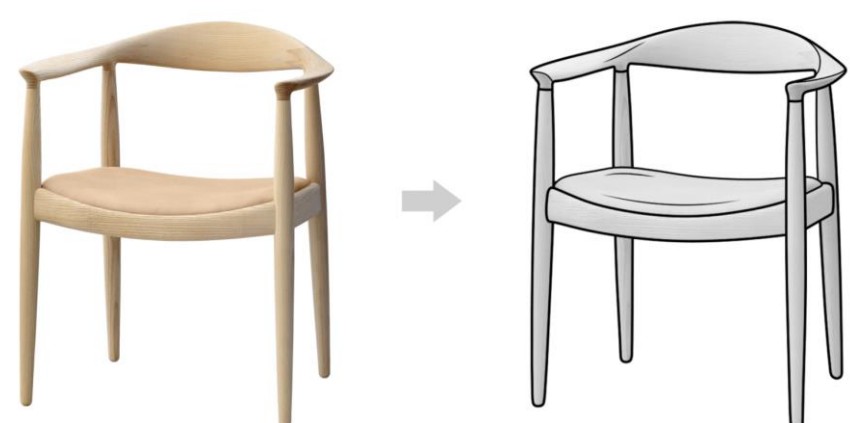

**Figure 6.** Representative samples converted into 2D images.

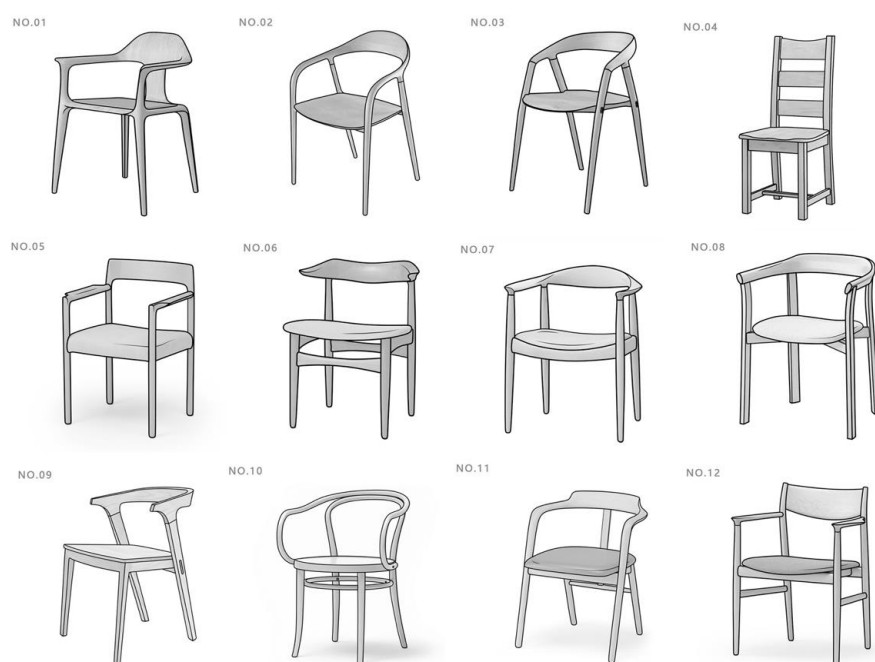

**Figure 7.** Twelve representative samples.

### 4.2. Results of the Modeling Imagery Adjectives Extraction and Factor Analysis

The classification of modeling imagery adjectives was carried out using a questionnaire approach with the help of experts. Thirty-two specialists were asked to participate in the experiment, including six literary instructors, twenty graduate students, and six Ph.D. students with design specialties. From a list of 140 adjectives, these participants were asked to choose 40–50 that best matched the modeling imagery descriptors of the solid wood chair concerning a two-dimensional schematic of a typical sample (selected by a multiscale approach). Lastly, based on the typical number of picks, the top 40 adjectives were chosen for further study (see Table 5).

**Table 5.** The 40 adjectives that were most recognized.

| | | | | |
|---|---|---|---|---|
| Quality | Artisanal | Detailed | Harmonious | Stylish |
| Crafted | Elegant | High-quality | Coordinated | Aligned |
| Simple | Durable | Overall | Eco-friendly | Lightweight |
| Practical | Safe | Sturdy | Novel | Balanced |
| Natural | Stable | Modern | Precise | Concordant |
| Refined | Superior | Demure | Orderly | Austere |
| Smooth | Unique | Traditional | Quaint | Delicate |
| Exquisite | Innovative | Serviceable | Plain | Comfortable |

The questionnaire study comprised 12 representative samples with the 40 most well-known descriptors. Convenience sampling was used to gather 122 questionnaires, of which 112 were valid (52 by males and 60 by females). Using SPSS, a first-factor analysis of the data was performed. After component analysis, 23 components with factor loadings above 0.6 in absolute value were chosen for the second-factor analysis, as shown in Table 6.

**Table 6.** The 23 adjective factors with an absolute load capacity value higher than 0.6.

| Adjectives | Initial | Extraction | Adjectives | Initial | Extraction |
|---|---|---|---|---|---|
| Harmonious | 1.000 | 0.782 | Safe | 1.000 | 0.761 |
| Coordinated | 1.000 | 0.765 | Serviceable | 1.000 | 0.671 |
| Orderly | 1.000 | 0.719 | Practical | 1.000 | 0.627 |
| Aligned | 1.000 | 0.703 | Crafted | 1.000 | 0.745 |
| Balanced | 1.000 | 0.686 | Quality | 1.000 | 0.718 |
| Concordant | 1.000 | 0.638 | Detailed | 1.000 | 0.717 |
| Unique | 1.000 | 0.774 | Exquisite | 1.000 | 0.601 |
| Novel | 1.000 | 0.768 | Traditional | 1.000 | 0.786 |
| Innovative | 1.000 | 0.745 | Plain | 1.000 | 0.637 |
| Stylish | 1.000 | 0.694 | Simple | 1.000 | 0.742 |
| Modern | 1.000 | 0.656 | Elegant | 1.000 | 0.718 |
| Stable | 1.000 | 0.805 | | | |

Factor analysis yielded a KMO value of 0.866, and Bartlett's test result was 3402.315 (df = 780, $p$ = 0.000), which was statistically significant. According to this finding, factor analysis may be used to identify standard components in the correlation matrix of the original cluster.

The transformed matrix showed significant differences in the five components derived from the second-factor analysis. After component analysis, factors with absolute values of factor loadings higher than 0.6 (16 factors in total) were selected, as shown in Table 7. Therefore, the following analysis can use all sixteen adjectives and five component factors from this factor analysis.

**Table 7.** The component matrices following the transformation.

| Adjectives | Component | | | | |
|---|---|---|---|---|---|
| | **1** | **2** | **3** | **4** | **5** |
| Aligned | 0.838 | | | | |
| Balanced | 0.812 | | | | |
| Concordant | 0.807 | | | | |
| Coordinated | 0.750 | | | | |
| Orderly | 0.727 | | | | |
| Unique | | 0.867 | | | |
| Novel | | 0.806 | | | |
| Innovative | | 0.800 | | | |
| Modern | | 0.669 | | | |
| Practical | | | 0.741 | | |
| Simple | | | 0.608 | | |
| Quality | | | | 0.786 | |
| Crafted | | | | 0.751 | |
| Detailed | | | | 0.632 | |
| Traditional | | | | | 0.860 |
| Plain | | | | | 0.638 |

By the second factor analysis, five groups consisting of sixteen adjectives were established and named "balanced and coordinated", "unique and novel", "practical and simple", "quality and detailed", and "traditional and plain" for further study, as shown in Table 8.

**Table 8.** Naming of each factor (groups of adjectives).

| Factor | Adjective Groups | Factor Naming | Code |
|---|---|---|---|
| 1 | Aligned, Balanced, Concordant, Coordinated, Orderly | Balanced and Coordinated | B & C |
| 2 | Unique, Novel, Innovative, Modern, Elegant | Unique and Novel | U & N |
| 3 | Practical, Simple, Serviceable | Practical and Simple | P & S |
| 4 | Quality, Crafted, Detailed | Quality and Detailed | Q & D |
| 5 | Traditional, Plain | Traditional and Plain | T & P |

*4.3. Results of Fuzzy Manipulation*

A modeling imagery evaluation questionnaire created by the seven-point Likert scale matching the seven-level fuzzy meaning, as shown in Table 2, could be derived from the twelve representative samples (two-dimensional images) shown in Figure 5 and the five renamed sets of adjective vocabularies in Table 8. A convenience selection technique was used to choose 120 individuals, and 117 valid questionnaires, including those from 51 men and 66 women, were obtained. They were between the ages of 18 and 40. A triangle membership function was used to quantify the fuzzy meaning of the survey responses as triangular fuzzy numbers. Table 9 displays their mean values after addition and averaging. As illustrated in Figure 8, the triangle blur maps were created after the participants evaluated the modeling images of the 12 exemplary samples.

**Table 9.** Ranking and mean of modeling imagery evaluation of 12 representative samples of solid wood chairs.

| Balanced and Coordinated | Unique and Novel | Practical and Simple | Quality and Detailed | Traditional and Plain |
|---|---|---|---|---|
| S7 (5.4, 7.1, 8.5) | S9 (5.2, 7.1, 8.5) | S1 (5.3, 7.1, 8.6) | S7 (5.1, 6.9, 8.4) | S7 (4.4, 6.2, 7.9) |
| S1 (4.7, 6.6, 8.2) | S10 (5.0, 6.8, 8.4) | S2 (5.0, 6.8, 8.4) | S1 (4.8, 6.5, 8.1) | S2 (4.1, 5.9, 7.6) |
| S11 (4.5, 6.4, 8.1) | S6 (4.4, 6.2, 7.9) | S7 (4.9, 6.7, 8.3) | S9 (4.3, 6.1, 7.8) | S1 (4.0, 5.8, 7.5) |
| S12 (4.5, 6.4, 8.1) | S7 (4.4, 6.2, 7.9) | S9 (4.6, 6.5, 8.1) | S11 (4.1, 6.0, 7.8) | S12 (3.9, 5.8, 7.6) |
| S9 (4.5, 6.3, 8.0) | S8 (4.3, 6.2, 7.9) | S8 (4.3, 6.2, 7.8) | S6 (4.1, 5.9, 7.6) | S11 (3.9, 5.7, 7.6) |
| S2 (4.4, 6.3, 8.0) | S1 (4.0, 5.8, 7.5) | S3 (4.2, 6.1, 7.8) | S8 (3.9, 5.8, 7.5) | S4 (4.0, 5.7, 7.2) |
| S8 (4.4, 6.2, 7.8) | S11 (3.7, 5.5, 7.3) | S11 (4.1, 6.0, 7.7) | S10 (3.9, 5.6, 7.3) | S3 (3.7, 5.6, 7.3) |
| S5 (4.4, 6.2, 7.7) | S3 (3.4, 5.2, 6.9) | S5 (3.9, 5.8, 7.5) | S3 (3.8, 5.6, 7.4) | S5 (3.9, 5.6, 7.4) |
| S10 (4.0, 5.7, 7.4) | S12 (3.4, 5.2, 6.9) | S6 (3.9, 5.8, 7.5) | S12 (3.8, 5.6, 7.4) | S9 (3.7, 5.5, 7.3) |
| S6 (3.9, 5.7, 7.4) | S2 (2.9, 4.6, 6.5) | S12 (3.8, 5.7, 7.5) | S2 (3.7, 5.6, 7.3) | S8 (3.6, 5.5, 7.2) |
| S3 (3.8, 5.6, 7.4) | S5 (2.8, 4.4, 6.2) | S10 (3.5, 5.2, 7.0) | S5 (3.3, 5.1, 6.8) | S6 (3.6, 5.4, 7.2) |
| S4 (3.8, 5.5, 7.1) | S4 (1.5, 2.8, 4.4) | S4 (3.3, 5.0, 6.7) | S4 (1.9, 3.5, 5.3) | S10 (3.4, 5.1, 6.8) |

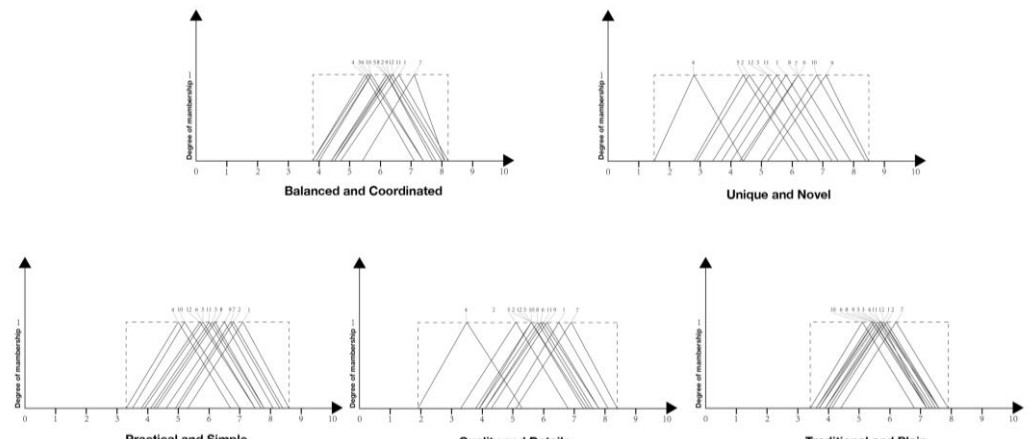

**Figure 8.** The triangular fuzzy numbers of the 12 representative samples in each modeling imagery evaluation.

Based on Table 9, the triangle fuzzy numbers of each sample were deblurred to produce the absolute utility values of the modeling imagery evaluation for 12 exemplary samples (see Table 10).

**Table 10.** Absolute utility values of 12 representative samples of solid wood chairs.

| Samples | Balanced and Coordinated | Unique and Novel | Practical and Simple | Quality and Detailed | Traditional and Plain |
|---|---|---|---|---|---|
| S1 | 0.5868 | 0.4754 | 0.6621 | 0.5787 | 0.4754 |
| S2 | 0.5454 | 0.3097 | 0.6195 | 0.4439 | 0.4900 |
| S3 | 0.4493 | 0.3878 | 0.5164 | 0.4493 | 0.4439 |
| S4 | 0.4305 | 0.0278 | 0.3603 | 0.1451 | 0.4570 |
| S5 | 0.5287 | 0.2775 | 0.4729 | 0.3732 | 0.4516 |
| S6 | 0.4608 | 0.5338 | 0.4729 | 0.4900 | 0.4203 |
| S7 | 0.6644 | 0.5232 | 0.6048 | 0.6324 | 0.5338 |
| S8 | 0.5313 | 0.5309 | 0.5284 | 0.4729 | 0.4294 |
| S9 | 0.5484 | 0.6566 | 0.5722 | 0.5192 | 0.4348 |
| S10 | 0.4724 | 0.6195 | 0.3932 | 0.4485 | 0.3750 |
| S11 | 0.5599 | 0.4348 | 0.5019 | 0.5046 | 0.4667 |
| S12 | 0.5599 | 0.3878 | 0.4614 | 0.4493 | 0.4758 |

The evaluation of the modeling imagery of the solid wood chairs was then ranked and charted with triangular fuzzy numbers (Figure 8). As Figure 8 shows, the different kinds

of solid wood chairs differ less in the visual evaluation of "balanced and coordinated", "practical and simple", and "traditional and plain", but are more differentiated in terms of "unique and novel" and "quality and detailed".

As shown in Table 9, S7 generally ranked high in all evaluations. It was particularly outstanding in the three aspects of "balanced and coordinated", "quality and detailed", and "traditional and plain", and better in the other two aspects. S1 scored the highest in "practical and simple" and was more average in "quality and detailed". S9 and S10 scored high in "unique and novel" and "balanced and coordinated".

As Figure 9 shows, the shape of Group 1 solid wood chairs generally conforms to "balanced and coordinated" and "practical and simple". Group 1 of solid wood chairs have a strong sense of the overall armrest backrest, smooth and beautiful lines of appearance; the connection of the chair surface and legs is relatively simple and practical, with no extra parts, making people feel that these are two chairs with excellent comfort.

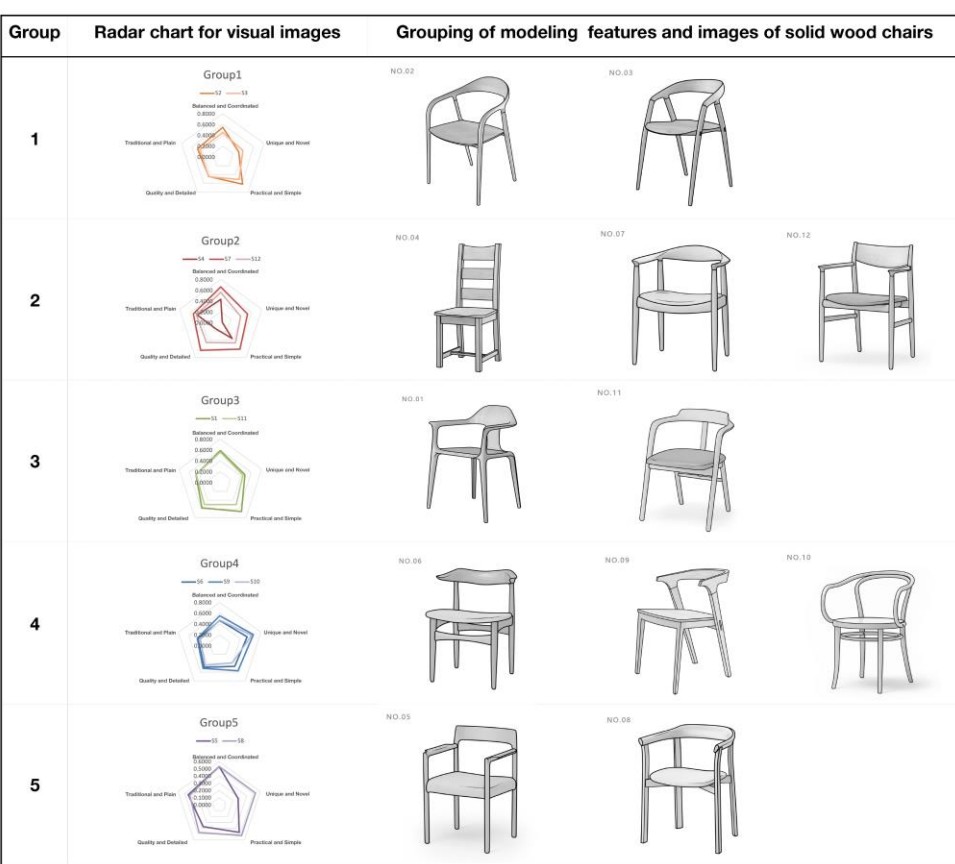

**Figure 9.** Five groups of solid wood chairs modeling images, with a similar comprehensive modeling imagery evaluation.

In Group 2, solid wood chairs are generally "balanced and coordinated", "traditional and plain", and "practical and simple". The solid wood chairs in Group 2 have the taste of account furniture and classical furniture. For Group 3, the components of the solid wood chair include a concealed connection, and the transition details are exceptionally delicate and of high quality. In Group 4, the solid wood chair styling is unique and novel in terms of visual recognition to a certain degree. In Group 5, in terms of the solid wood chair shape, the proportional relationship between the components is more balanced and coordinated, and a soft cushion design can increase the comfort of the user sitting.

As can be seen from Figure 9, S2 and S3 are very close to the modeling imagery evaluation value, scoring high in "practical and simple" and "traditional and plain". However, they do not perform as well in terms of "unique and novel" and "quality and detailed".

S4, S7, and S12 have similarities in terms of "traditional and plain" and "practical and simple". S7 ranks first in terms of "balanced and coordinated", "quality and detailed", and "traditional and plain", and is first in terms of "unique and novel" and "practical and simple". S1 and S11 are very close to each other in modeling imagery evaluation and score high in terms of "practical and simple" and "quality and detailed". S6, S9, and S10 are rich in styling variations, and S10 has especially elegant curves and visual aesthetics. S5 and S8 have some similarities in terms of "balanced and coordinated", "traditional and plain", and "practical and simple". According to the questionnaire, these two types of solid wood chairs are in the middle ranking in all aspects. Respondents generally believe the chair surface is a soft package design and should be comfortable for sitting.

*4.4. Experimental Validation*

This experiment used a semi-structured interview method to investigate consumers' preferences and emotional experiences with the modeling imagery of representative solid wood chairs (as shown in Figure 10). The interview results showed that the "traditional and plain" and "balanced and coordinated" solid wood chairs were suitable for placing in a tea space, with an antique and calm atmosphere that would show the cultural sophistication of the user. The "practical and simple" and "quality and detailed" solid wood chairs, with a better sense of comfort and taste due to their simple shape and exquisite craftsmanship, were suitable for modern minimalist spaces, which are simple and elegant and give users a naturally warm and pleasant feeling. The "unique and novel" solid wood chairs were more suitable for aesthetic spaces, cafes, and restaurants due to their rich colors and stylish shapes, and contemporary young people loved their distinctive forms. A comprehensive understanding of the furniture style preferences of different consumer groups provides a reference for consumers to purchase furniture and designers in the furniture styling stage.

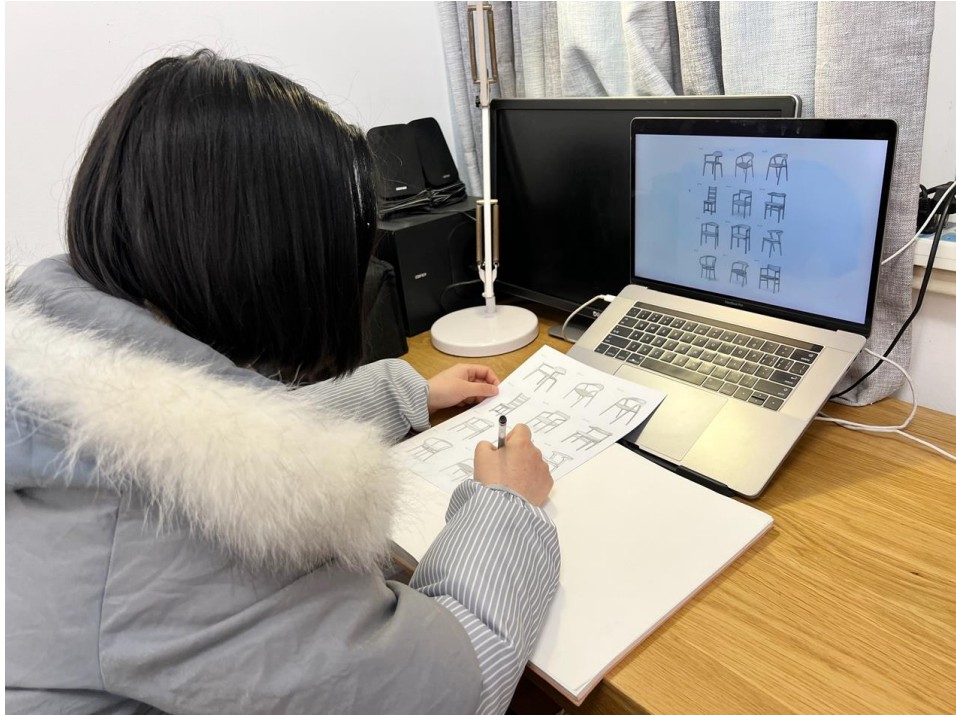

**Figure 10.** The participant was in a semi-structured interview.

**5. Conclusions and Suggestions**

Based on Kansei engineering, this study uses an integrated approach of MDS, cluster analysis, an expert questionnaire, factor analysis, fuzzy theory, and semi-structured interviews to investigate the evaluation of modeling imagery of solid wood chairs.

In terms of theoretical significance, this study contributes to the study of the modeling of solid wood chairs. It provides a reference model for researchers to conduct such studies. Since the accuracy of the evaluation sample and modeling imagery adjectives further affect the objectivity and accuracy of the study results, we used MDS and cluster analysis to sort out the representative sample at the beginning; then, the imagery adjectives are effectively selected through expert questionnaires and factor analysis. Finally, the evaluation of modeling imagery of solid wood chairs is further analyzed using the triangular fuzzy number operation in fuzzy theory. The sample selection method in this study is a scientific technique that differs from previous studies because the samples are transformed into two-dimensional images to avoid confounding factors, such as angles, colors, textures, and brand preferences. The study not only unveils the composition of solid wood chairs (as shown in Figure 5), but also provides designers with structural and detailed references when designing chairs. Moreover, the 123 images of solid wood chairs collected (shown in Figure 4) can form a database of solid wood chairs so that companies and designers can use this study's results.

In terms of practical significance, participants' impressions, psychological feelings, and favorite preferences on the modeling of solid wood chairs could be understood. A total of twelve representative samples and five sets of adjectives were obtained. The results show that the different modeling of the solid wood chairs presents unique visual effects. A total of 12 representative samples differ in the modeling imagery evaluation of "balanced and coordinated", "practical and simple", and "traditional and plain". However, it is more differentiated in terms of "unique and novel" and "quality and detailed". There are overall similarities in some samples; for example, for S1 and S11, some assessments are similar, and furniture manufacturers and designers may consider alternatives to both. Aesthetic research on modeling solid wood chairs can help furniture brands and designers improve and create modeling attributes of new products to meet current differentiated and precise market demands, respond in a timely manner, and create differentiated marketing strategies. Thus, the results of this study reveal the influence of modeling attributes of solid wood chairs on consumer perceptions.

## 6. Research Limitations and Future Directions

The findings of this study are generalizable but have certain restrictions. First, this study aims to model solid wood chairs, consumer assessment elements, and favorite preferences to adjust with confounders. In order to investigate their differences in the future, various materials, colors, brands, or styles should be employed as study variables.

Second, most of the study's participants were from the Chinese region and had a common cultural background. The results only apply to the locals because of differences in terms of lifestyle, culture, and geography; they may not be an appropriate fit for those in another area. Different cultural backgrounds may have different emotional affections and aesthetic views.

In addition, most of the study's participants were young and middle-aged; therefore, more generalizability of the results is required. To make the results more applicable across genders, age groups, and vocations, further research must be carried out in the future.

Finally, the study's samples were in the form of photographs, which may not have given the participants the most intuitive impressions. In the future, representative samples of solid wood chairs might be made into survey entities, enabling participants to touch the natural materials as closely as possible for their evaluations, thus improving the study's accuracy.

This study is the first series on consumers' perception of solid wood chair modeling. The need for study on this subject is expanding due to the increasing demand for quality of living and the improvement in aesthetic consciousness.

**Author Contributions:** Conceptualization, L.X. and Y.P.; methodology, L.X. software, L.X.; validation, L.X.; formal analysis, L.X.; investigation, L.X.; data curation, L.X.; writing—original draft preparation, L.X.; writing—review and editing, L.X.; visualization, L.X.; supervision, Y.P. All authors have read and agreed to the published version of the manuscript.

**Funding:** This research received no external funding.

**Institutional Review Board Statement:** Not applicable.

**Informed Consent Statement:** Informed consent was obtained from all subjects involved in the study.

**Data Availability Statement:** Not applicable.

**Acknowledgments:** Thanks to the anonymous reviewers for their comments and efforts to help improve the paper.

**Conflicts of Interest:** The authors declare no conflict of interest.

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
