# Peer review of "Study on Imagery Modeling of Solid Wood Chairs in Big Data"

_electronics, doi:10.3390/electronics12081949_

Round 1
Reviewer 1 Report
The authors evaluated how the modelling imagery of solid wood chairs affects user preferences and emotional experiences. They obtained a representative sample by Multidimensional Scaling (MDS) and factor analysis, and five groups of adjective vocabulary were selected to describe the modelling imagery. The Triangular Fuzzy Theory was applied to analyse and discuss the 12 types. They verified that the differences in evaluations of the 12 samples in the "unique and novel" and "quality and detailed" groups were significant, but more minor differences in the groups of "traditional and plain", "balanced and coordinated" and "practical and simple". Five groups with similar modelling imagery were created, and solid wood chairs with different modelling imagery were placed in suitable spaces. They found that the evaluation of the modelling imagery of solid wood chairs can be scientifically and effectively used to accurately reflect the perception needs of consumers and improve the design efficiency of the furniture product development stage.
I put my observations as under:
1. The study takes a while. It is necessary to squeeze every area.
2. There is no literature review. It is desperately needed to discuss how you came up with this study. Who motivated you to choose/use such and such algorithms and methods to solve the current problem? For instance, tell us about your training in MDS, cluster analysis, expert questionnaire, factor analysis, fuzzy 463 theory, and semi-structured interviews. Before you tell me how you solve this problem, readers must be aware of the state-of-the-art.
3. Limitations of the study are absent.
4. Directions for future research should be given.
Author Response
Dear Reviewer:
Re: Manuscript ID: 2299072 and Title: Study on Imagery Modeling of Solid Wood Chairs in Big Data
Thank you for your letter and the reviewer's comments concerning our manuscript, "Study on Imagery Modeling of Solid Wood Chairs in Big Data " (2299072). Those comments are valuable and very helpful. We have read through the comments carefully and made corrections. Based on the instructions provided in your letter, we uploaded the file of the revised manuscript. Revisions in the text are highlighted and presented below; please consult the manuscript file for details.
We'd like to thank you for letting us send you a revised copy of the manuscript again, and we really appreciate your time and thought.
Sincerely yours,
Corresponding Author:
Younghuan Pan,
Department of Smart Experience Design,
Kookmin University, Seoul, 02707, Korea
Tel: +886-933-482656
E-mail: peterpan@kookmin.ac.kr
Reviewer #1:
Q1. The study takes a while. It is necessary to squeeze every area.
Response: We are grateful for the suggestion. I have compressed some parts of the article and I am not sure if I understand what you mean. It is specifically reflected in the manuscript.
Q2. There is no literature review. It is desperately needed to discuss how you came up with this study. Who motivated you to choose/use such and such algorithms and methods to solve the current problem? For instance, tell us about your training in MDS, cluster analysis, expert questionnaire, factor analysis, fuzzy theory, and semi-structured interviews. Before you tell me how you solve this problem, readers must be aware of the state-of-the-art.
Response: We are grateful for the suggestion. We have added a literature review section with the corresponding research methods and theories. Please refer to the manuscript for details.
Q3. Limitations of the study are absent.
Response: We are grateful for the suggestion. We have added a sixth section: limitations of the study and future directions. Please refer to the manuscript for details.
Q4. Directions for future research should be given.
Response: We are grateful for the suggestion. We have added a sixth section: Limitations of the study and future directions. Please refer to the manuscript for details.
Please see the attachment!Thank you so much! Good luck with your work and enjoy your life!

Reviewer 2 Report
Dear Authors,
the manuscript is very interesting. The research into imagery modeling of solid wood chairs in big data is excellenty written.
Some suggestions:
Many research studies have been focused on the calculation and experimental testing of chair construction since 1980, authors investigated mostly calculations associated with chairs, weak points of joints, load limit capacity (for example in case of users with higher body weight), stiffness, etc. Many articles focused also on FEM or direct stiffness analysis. All of these results strongly influence the shape and properties of the design of the chair and authors of this manuscript should add at least some discussion to this problem and connect it with the evaluation of how the modeling imagery of solid wood chairs affects user preferences and emotional experiences. Please check some sources here:
Hajdarević, S., & Busuladžić, I. (2015). Stiffness analysis of wood chair frame. Procedia Engineering, 100, 746-755.
Langová, N., Réh, R., Igaz, R., Krišťák, Ľ., Hitka, M., & Joščák, P. (2019). Construction of wood-based lamella for increased load on seating furniture. Forests, 10(6), 525.
Haviarova, E., Eckelman, C., & Erdil, Y. (2001). Design and testing of environmentally friendly wood school chairs for developing countries. Forest products journal, 51(3), 58-64.
Chen, B., Yu, X., & Hu, W. (2022). Experimental and numerical studies on the cantilevered leg joint and its reinforced version commonly used in modern wood furniture. BioResources, 17(3), 3952.
Hitka, M., Joščák, P., Langová, N., Krišťák, L., & Blašková, S. (2018). Load-carrying capacity and the size of chair joints determined for users with a higher body weight. Bioresources, 13(3), 6428-6443.
Güray, E., Ceylan, E., & Kasal, A. (2022). Weight-strength optimization of wooden household chairs based on member section size. Maderas. Ciencia y tecnología, 24.
Paoliello, C., & Mantilla Carrasco, E. V. (2008). Chair load analysis during daily sitting activities. Forest products journal, 58(9).
Shah, R. M., Bhuiyan, M. A. U., Debnath, R., Iqbal, M., & Shamsuzzoha, A. (2013). Ergonomics issues in furniture design: a case of a tabloid chair design. In Advances in Sustainable and Competitive Manufacturing Systems: 23rd International Conference on Flexible Automation & Intelligent Manufacturing (pp. 91-103). Springer International Publishing.
Cranz, G. (2000). The Alexander Technique in the world of design: posture and the common chair: Part I: the chair as health hazard. Journal of bodywork and movement therapies, 4(2), 90-98.
and many others.
In the Discussion section I suggest adding more discussion with previous research dealing with imagery modeling of furniture, influence of product design to the market and add Implications for the Market and further research.
Please revise the Conclusions part, it is not needed to repeat the results, please focus more on the Implications from your research.
Author Response
Dear Reviewer:
Re: Manuscript ID: 2299072 and Title: Study on Imagery Modeling of Solid Wood Chairs in Big Data
Thank you for your letter and the reviewer's comments concerning our manuscript, "Study on Imagery Modeling of Solid Wood Chairs in Big Data " (2299072). Those comments are valuable and very helpful. We have read through the comments carefully and made corrections. Based on the instructions provided in your letter, we uploaded the file of the revised manuscript. Revisions in the text are highlighted and presented below; please consult the manuscript file for details.
We'd like to thank you for letting us send you a revised copy of the manuscript again, and we really appreciate your time and thought.
Sincerely yours,
Corresponding Author:
Younghuan Pan,
Department of Smart Experience Design,
Kookmin University, Seoul, 02707, Korea
Tel: +886-933-482656
E-mail: peterpan@kookmin.ac.kr
Reviewer #2:
Q1. Many research studies have been focused on the calculation and experimental testing of chair construction since 1980, authors investigated mostly calculations associated with chairs, weak points of joints, load limit capacity (for example in case of users with higher body weight), stiffness, etc. Many articles focused also on FEM or direct stiffness analysis. All of these results strongly influence the shape and properties of the design of the chair and authors of this manuscript should add at least some discussion to this problem and connect it with the evaluation of how the modeling imagery of solid wood chairs affects user preferences and emotional experiences.
Response: We are grateful for the suggestion. We have reviewed the references you provided in detail and have added this section of the literature to explore and analyze previous questions about the structure, load-bearing capacity, and strength of the chair. Please refer to the manuscript for details.
Q2. In the Discussion section I suggest adding more discussion with previous research dealing with imagery modeling of furniture, influence of product design to the market and add Implications for the Market and further research.
Response: We are grateful for the suggestion. We have looked in detail at the references you provided and have added this section to the literature exploring the stylistic imagery of furniture, adding that the impact of solid wood chair stylistic imagery on the market and consumers has also been studied. Please refer to the manuscript for details.
Q3. Please revise the Conclusions part, it is not needed to repeat the results, please focus more on the Implications from your research.
Response: We are grateful for the suggestion. We have abridged the concluding section for repetitive content and elaborated on the value and implications of the study. Please refer to the manuscript for details.
Please see the attachment!Thank you so much! Good luck with your work and enjoy your life!

Round 2
Reviewer 1 Report
Thank you for addressing my concerns and modifying the article accordingly. I'm satisfied.
Author Response
Dear Reviewer:
Re: Manuscript ID: 2299072 and Title: Study on Imagery Modeling of Solid Wood Chairs in Big Data Data
Thank you for your letter and comments concerning our manuscript, And thanks for your approval of our revisions!
Good luck with your work, and have a happy life!
Sincerely yours,
Corresponding Author:
Younghuan Pan.
Department of Smart Experience Design,
Kookmin University, Seoul 02707, Korea
Tel: +886-933-482656
E-mail: peterpan@kookmin.ac.kr
F-Tel: +886-933-482656
E-mail: peterpan@kookmin.ac.kr